# A New Molecular Detection System for Canine Distemper Virus Based on a Double-Check Strategy

**DOI:** 10.3390/v13081632

**Published:** 2021-08-18

**Authors:** Sabrina Halecker, Sabine Bock, Martin Beer, Bernd Hoffmann

**Affiliations:** 1Institute of Diagnostic Virology, Friedrich-Loeffler-Institut, Federal Research Institute for Animal Health Südufer 10, 17493 Greifswald-Insel Riems, Germany; Sabrina.Halecker@fu-berlin.de (S.H.); martin.beer@fli.de (M.B.); 2Landeslabor Berlin-Brandenburg, Gerhard-Neumann-Str. 2, 15236 Frankfurt (Oder), Germany; Sabine.Bock@Landeslabor-bbb.de

**Keywords:** canine distemper, *Canine morbillivirus*, diagnostic, RT-qPCR, domestic animal, wildlife population, double-check strategy

## Abstract

Due to changing distemper issues worldwide and to inadequate results of an inter-laboratory study in Germany, it seems sensible to adapt and optimize the diagnostic methods for the detection of the canine distemper virus (CDV) to the new genetic diversity of virus strains. The goal of the project was the development, establishment and validation of two independent one-step reverse transcriptase quantitative polymerase chain reaction (RT-qPCR) methods for the safe detection of CDV in domestic and wild animals. For this purpose, an existing CDV-RT-qPCR was decisively adapted and, in addition, a completely new system was developed. Both CDV-RT-qPCR systems are characterized by a very high, comparable analytical and diagnostic sensitivity and specificity and can be mutually combined with inhibition or extraction controls. The reduction in the master mix used allows for the parallel implementation of both CDV-RT-qPCR systems without significant cost increases. For validation of the new CDV-RT-qPCR duplex assays, a panel comprising 378 samples derived from Germany, several European countries and one African country were tested. A sensitivity of 98.9% and a specificity of 100% were computed for the new assays, thus being a reliable molecular diagnostic tool for the detection of CDV in domestic and wild animals.

## 1. Introduction

The canine distemper virus (CDV) causes a severe, highly contagious infectious disease, affecting mainly carnivores [1,2,3]. CDV (taxonomic name: *Canine morbillivirus*) was first mentioned in 1760 and belongs to the genus *Morbillivirus* in the family *Paramyxoviridae* of the order *Mononegavirales* [4,5]. The genome of CDV encodes for eight proteins designated as nucleocapsid (N), phospho- (P), matrix (M), fusion (F), hemagglutinin (H), large polymerase (L) (structural proteins), and C and V proteins (nonstructural proteins) [6]. Serologically, the virus is classified into one serotype and distinct lineages that are genetically determined by sequence differences in the *H gene* [7]. According to their geographical distribution, the CDV lineages are clustered currently as America 1 to 4, South-America 1 to 3, Africa 1 to 2, Asia 1 to 2, Europe, European Wildlife, Arctic-like and Rockborn-like. In Europe, CDV isolates of the lineages Europe, European Wildlife and Arctic-like are predominant [3,6,8,9].

In recent years, an increase in the incidence of distemper disease in the global dog population as well as massive outbreaks in global wildlife were recorded [3,10,11,12]. Additionally, in European wildlife, outbreaks of distemper infections were reported progressively [13,14,15,16,17,18,19]. Overall, the vaccine fatigue of animal owners and the failure of established vaccines in connection with new virus variants are considered the cause of the rise in distemper cases [2,10,20,21,22,23]. A spillover of CDV from the dog to wild population and vice versa is suspected and poses a serious threat to endangered species [12,18,23,24,25,26].

Concerns about the increasing number of cases of canine distemper virus indicate the importance of targeted surveillance of CDV in susceptible species [17] and the need to reconsider established diagnostic methods for the detection of CDV. An inter-laboratory proficiency test conducted in Germany revealed standout deficiencies in the detection of certain CDV sequences detected by CDV reverse transcriptase quantitative polymerase chain reaction (RT-qPCR) assays commonly used. Consequently, a sequence alignment of CDV sequences from an international database (National Center for Biotechnology Information, NCBI) with the primer-probe sequences of established RT-qPCR assays was carried out. The results indicated the drawbacks of the RT-qPCR systems currently in use; thus, we strived for an adaption to the new virus variants. Commonly used RT-qPCR systems for the detection of CDV with relevance to this study are based on the primer-probe mixes described by Elia and co-workers and by Scagliarini and co-workers [27,28]. The ELIA assay is located on the *N gene*, and the analytical sensitivity was determined as 10^2^ genome equivalent (GE)/μL, while the primer-probe mix of the Scagliarini assay targeting the P protein demonstrated a limit of detection (LOD) of 10^1^ GE/μL [27,28].

In this study, an established CDV-RT-qPCR assay designed by Scagliarini and co-workers was decisively adapted and a new CDV-RT-qPCR assay was designed based on the latest available CDV sequence information. To ensure a safe and sensitive detection of broad-range CDV sequences, the new diagnostic approach is targeted on a double-check strategy as described previously for other pathogens [29,30]. Both CDV-RT-qPCR assays were equipped with internal controls (IC) and validated with a comprehensive validation panel comprising 378 samples, where 180 specimens tested negative and 198 positive for CDV.

## 2. Materials and Methods

### 2.1. Samples

For this study, a total of 378 animal samples were available from routine diagnostics including 198 CDV-positive (52.4%) and 180 negative samples (47.6%) that were predetermined by an established, routinely used CDV-specific RT-qPCR assay described by Elia and co-workers [27]. The majority of sampling was carried out on domestic and wild animals in the German federal states Hesse (n = 139), Brandenburg (n = 2), Berlin (n = 1), Lower Saxony (n = 1), Saxony Anhalt (n = 1) and Thuringia (n = 1). The panel is also comprised of specimens derived from Austrian (n = 8), Italian (n = 1), Spanish (n = 1), Romanian (n = 1), Ukrainian (n = 1) and African (n = 1) animals, allowing for the inclusion of a wider geographical distribution. Among the animal specimens, 47% were collected from foxes (n = 179) and 29% were collected from racoons (n = 111), while further sample material is represented by martens (n = 23), dogs (n = 18), racoon dogs (n = 8), wolves (n = 8), badgers (n = 6), minks (n = 6), polar fox (n = 1), otter (n = 1), red panda (n = 1) and without specification (n = 11). More than one tissue sample was available from some animals.

Additionally, a log_10_ dilution series (10^−1^ to 10^−8^) of the CDV strain “Onderstepoort” (sample ident number: BH 04/18-2) was used and a small panel of phocine distemper virus (PDV)-positive samples (n = 4) was also tested.

### 2.2. CDV-RT-qPCR Assays

To improve the reliability of canine distemper diagnosis, two CDV-RT-qPCRs were designed for simultaneous application. According to the NCBI database, an existing RT-qPCR assay previously published by Scagliarini and co-workers [28] and targeting the *P gene* was decisively adapted to the latest available CDV sequence information. A second, completely new RT-qPCR was selected from an initial set of 12 varying primer-probe combinations (CDV-Mix 4 to CDV-Mix 15) that were located on heterogeneous genetic locations (*N*, *P* and *L gene*) within the CDV genome. The primers and TaqMan-probes were designed based on an alignment of 2075 CDV sequence information available from the NCBI database.

To further enhance the security of the new detection systems, both CDV-RT-qPCR assays were carried out in parallel with an optimized total reaction volume of 12.5 μL per approach and were additionally equipped with two different IC systems (double-check strategy). In this study, the adapted assay by Scagliarini and co-workers, hereinafter designated as CDV-Mix 3, was combined with an IC based on heterologous RNA (EGFP-Mix 2-HEX) [31] while the newly established RT-qPCR assay was used in combination with an endogenous IC (ß-Actin-Mix 2-HEX) [32] (Table 1).

All protocols were carried out by using the AgPath-ID™ One-Step RT-PCR Reagents (Thermo FisherScientific, Waltham, MA, USA) in a 12.5 µL approach. The master mix was prepared with 1.25 µL RNase free water, 6.25 µL 2× RT-PCR buffer, 0.5 µL 25× RT-PCR Enzyme mix, 1.0 µL primer-probe mix specific for the CDV detection, 1.0 μL primer-probe mix for the IC system (EGFP-Mix 2-HEX or ß-Actin-Mix 2-HEX) and a 2.5 µL template. For generation of the CDV primer-probe-mixes, 10 µL each of the forward and reverse primer (100 pmol/µL), 2.5 µL of the probe (100 pmol/µL) and 77.5 µL 0.1× Tris-ethylenediaminetetraacetic acid (EDTA) (pH = 8.0) were combined. For the IC primer-probe-mixes, 2.5 µL each of the forward and reverse primer (100 pmol/µL), 2.5 µL of the probe (100 pmol/µL) and 92.5 µL 0.1× Tris-EDTA (pH = 8.0) were mixed. The temperature-time profile was performed on the Bio-Rad CFX96™ Real-Time PCR Detection System (BioRad Laboratories Inc., Hercules, CA, USA). A reverse transcriptase at 45 °C for 10 min, an activation step at 95 °C for 10 min with a subsequent cycling process (n = 45) of denaturation at 95 °C in 15 s, annealing at 56 °C in 20 s and an elongation at 72 °C in 30 s were realized.

### 2.3. Generation of a CDV Standard

For the determination of the method’s analytical sensitivity, a CDV standard was produced on the basis of a whole virus standard, of which the GE copies per µL were determined by droplet digital PCR (ddPCR). For this purpose, a log_10_ dilutional series (10^−2^ to 10^−4^) of the strain “Onderstepoort” (sample ident number: BH 04/18-2) was prepared and tested in duplicate on the QX200 Droplet Digital PCR system (Bio-Rad Laboratories Inc., Hercules, CA, USA) according to the manufacturer’s recommendations by using the One-Step RT-ddPCR Advanced Kit for Probes (Bio-Rad Laboratories Inc., Hercules, CA, USA). The master mix contained 5.0 µL RNase free water, 5.0 µL Supermix, 2.0 µL Reverse Transcriptase, 1.0 µL 300 mM DTT, 2.0 µL CDV-Mix 3 and a 5.0 µL template. The results of the ddPCR were analyzed by using the software “QuantaSoft™ Analysis Pro Software“ (Bio-Rad Laboratories Inc., Hercules, CA, USA). The prerequisite for the absolute quantification of the CDV standard was a clear discrimination of negative and positive droplets generated during the ddPCR run; thus, the absolute quantification of the viral standard (in GE copies per µL) was determined.

### 2.4. Analytical Sensitivity

Three independent biological replicates were prepared, each comprising a six-fold log_10_ dilutional series (10^4^ to 10^−1^) of the whole virus standard diluted in a suspension of Madin Darby Canine Kidney (MDCK) cells. Each biological replicate was separately applied with the two selected CDV-RT-qPCRs on four different days containing three replicates per day, thus resulting in a total of 12 technical replicates per assay. The LOD was determined as the virus titer in GE copies per μL that positively detects 95% of the replicates by the selected PCR assay [33,34]. Calculations of standard deviation (SD) and coefficient of variation (CV) were performed according to Vandemeulebroucke and co-workers [35].

### 2.5. Analytical Specificity

The inclusivity (detection of samples containing related target organisms) and exclusivity (detection of only the unique target organism, but no cross-reaction to related target organisms) [36] of the CDV-specific RT-qPCR assays were tested on several available CDV strains (n = 15) and on a panel of morbillivirus representatives comprising peste des petits ruminants virus (PPRV, n = 5), phocine distemper virus (PDV, n = 4), porpoise morbillivirus (PMV, n = 1), dolphin morbillivirus (DMV, n = 1) and the measles virus (MV, n = 2). A panel of paramyxovirus representatives comprising Nipah virus (n = 1), Hendra virus (n = 1), Newcastle disease virus (NDV, n = 1) and bovine parainfluenza virus 3 (BPIV-3, n = 1) were also included in the analytical testing.

### 2.6. Diagnostic Sensitivity and Specificity

For further validation of the methods regarding diagnostic sensitivity and specificity, a comprehensive sample panel (see Section 2.1) containing CDV-positive specimens (n = 198) and negative specimens (n = 180) was used. All samples of the validation panel were tested, with the two CDV-RT-qPCR assays currently being validated plus the CDV-RT-qPCR assay of Elia and co-workers used as a reference method [27]. Diagnostic sensitivity and specificity were calculated in accordance with the two-by-two table [37,38].

### 2.7. Receiver Operating Curve (ROC) Analyses

ROC analyses were performed to determine the accuracy of the applied CDV-RT-qPCR detection systems and for a comparative evaluation between the different primer-probe assays. The test results grouped as positive and negative outcomes were contrasted, and sensitivity and 1-specificity were calculated and depicted as an ROC diagram. ROC analyses were performed according to Akobeng [39].

### 2.8. Quality Assessment and Bias Statement

All studies were carried out following the Office International des Epizooties (OIE) manual *Principles and Methods of Validation of Diagnostic Assays for Infectious Diseases* in an appropriate and reliable manner, as it is the industry standard for the development of veterinary diagnostic detection systems [36].

However, to standardize the parameters addressed for bias assessment, further criteria were taken into consideration [40,41]. These included conducting all analyses by one operator, independently interpreting the test results from an index test and reference standard and ensuring that an appropriate interval between index test and reference standards were contemplated to guarantee best bias control.

## 3. Results

In total, 13 CDV-specific primer-probe mixes were designed and tested comparatively. A first test series carried out with a log_10_ dilutional series (10^−1^ to 10^−8^) of the CDV strain “Onderstepoort” (sample ident number: BH 04/18-2) lead to the preselection of the CDV-Mixes 3, 5, 7, 9 and 15 (Appendix A), considering the sensitivities and fluorescence data of all 13 CDV mixes. A diverse localization within the CDV genome was also taken into account. In terms of sensitivities, both the number of detectable dilution stages and preferably the highest relative fluorescence units (RFUs) on the last dilution stage were assessed by the CDV-Mix taken into account. For CDV-Mixes revealing similar results in both of these categories, a decision was made according to their location on the CDV genome. Further test series based on the exclusivity and inclusivity (Appendix A) of the CDV-specific RT-qPCR assays displayed discrepancies for CDV-Mix 9 and 15 due to their non-specific detection pattern. According to their location on the genome, their sensitivity and their detectable RFUs, CDV-Mix 3 and CDV-Mix 7 proved to be the most appropriate CDV-RT-qPCR assays, ensuring a clear discrimination between positive and negative test results (Appendix A). This was also reinforced by an in silico testing of the described oligonucleotides. The primers that were ultimately used and the TaqMan-probes for the CDV detection and the IC systems are summarized in Table 1.

Concerning the LOD, both CDV-RT-qPCR assays displayed a similar analytical sensitivity of less than 10^1^ GE/μL template. When using CDV-Mix 3, 31 of 36 replicates containing a template concentration of 10^1^ GE/μL scored positive, while only 13 replicates showed positive results by the application of CDV-Mix 7 (Table 2), which indicated a slightly better analytical performance for CDV-Mix 3. Standard deviations (SDs) of the quantification cycle (Cq)-values ranged from 0.14 to 1.11 for CDV-Mix 3 and from 0.17 to 1.15 for CDV-Mix 7. Interestingly, the highest SDs with numerical values of more than 1.0 were only achieved with replicates containing a template concentration of 10^1^ GE/μL, regardless of the CDV-RT-qPCR assay used (Table 2).

In terms of analytical specificity, CDV-Mix 3 and CDV-Mix 7 were able to discriminate between the target “CDV-RNA” and the RNA of other related viruses, as 15 CDV strains were tested positive by both RT-qPCR assays. However, all specimens containing related representatives of the genus *Morbillivirus* (PPRV, PDV, PMV, DMV and MV) or the family *Paramyxoviridae* (Nipah virus, Hendra virus, NDV and BPIV-3) species were tested negative by both CDV-RT-qPCR assays.

Regarding diagnostic sensitivity and specificity, the selected CDV-RT-qPCR assays displayed comparable results when compared with the results received from the ELIA assay. All negative samples tested negative when performed with CDV-Mix 3 and CDV-Mix 7, indicating a diagnostic specificity of 100%. For the positive sample panel, only 196 of 198 specimens scored positive in both duplex-PCR systems (Appendix A). The diagnostic sensitivity for the new CDV-RT-qPCR assays was calculated as 98.9%. Within all applied PCR runs, functionality of the integrated heterologous and endogenous ICs was verified and stated.

The diagnostic accuracy of the CDV-RT-qPCR assays was determined by calculating and visualizing the ROCs (Figure 1). Based on the ROC, the area under the curve (AUC) was calculated. The CDV-Mix 3 revealed an AUC of 0.960 (95% Confidence Interval (CI): 0.903 to 1.000). The AUC of CDV-Mix 7 was 0.970 (95% CI: 0.926 to 1.000), and that of the ELIA-Mix was 0.966 (95% CI: 0.913 to 1.000). According to these results, the three CDV-RT-qPCR assays had very high diagnostic accuracies.

## 4. Discussion

Global considerations about increasing case reports of CDV reinforced concerns about suspected new virus variants, which also indicate a challenge for established diagnostic methods for the molecular detection of CDV. An inter-laboratory proficiency test in Germany and a sequence alignment with the latest available CDV sequences on the NCBI database and sequences of established CDV-specific primer-probe mixes revealed the diagnostic incompatibilities of currently used CDV-RT-qPCR assays. Moreover, in this study, a small panel of PDV-positive samples (n = 4) were tested by the comparative application of CDV-Mix 3, CDV-Mix 7 and the ELIA-Mix. Here, the ELIA-Mix displayed obvious drawbacks regarding its analytical exclusivity as two of four PDV-positive samples were recognized as positive. In the test series, which was carried out for validation purposes of the new CDV-RT-qPCR, the ELIA-Mix revealed major discrepancies in the sensitive detection of a few isolates concerning the Cq values (Appendix A). Regarding the publication and validation of the ELIA-Mix, a reliable and transparent IC system was also unable to be established. These major findings confirmed standout deficiencies concerning the CDV-RT-qPCR currently in use and support the necessity of a revision of CDV-specific RT-qPCRs. In order to advance safe and reliable CDV diagnosis in domestic and wild animals, an extended molecular diagnostic approach based on RT-qPCR technology with double-check strategy was developed.

A decisively modified Scagliarini assay [28], which were designated as CDV-Mix 3 and located on a conserved region of the *P gene*, were combined with a newly designed RT-qPCR assay referred to as CDV-Mix 7 and targeting the *P gene*, These two independent CDV-RT-qPCR assays were applied for the double-check PCR system. The new PCR assays were proven to be highly sensitive and specific RT-qPCR systems that were adapted to the latest available CDV sequence information and, thus, met the requirements of improved CDV diagnostic methods. The analytical sensitivity of both new CDV-RT-qPCR assays was similar, as a copy number of less than 10^1^ GE/μL was set as the LOD for both assays. When considering the LODs of the original Scagliarini assay (10^1^ GE/μL) and the ELIA assay (10^2^ GE/μL), the newly validated RT-qPCR assays proved to be highly sensitive in terms of analytical performance. In addition, the discrimination of non-CDV species was analyzed in an initial test series, as deficiencies were stated for established CDV-RT-qPCRs. CDV-Mix 3 and CDV-Mix 7 consistently detected positive samples of virus species belonging to the genus *Morbillivirus* and/or the family *Paramyxoviridae* and those not CDV-positive were denoted as negative; thus, the new CDV-RT-qPCR assays were demonstrated to be superior to the ELIA-Mix in terms of analytical specificity. Comprehensive validation test series were performed on a panel of 378 samples and revealed a diagnostic sensitivity of 98.9% and a diagnostic specificity of 100% for both assays, including the reliable testing of samples belonging to different host species (foxes, racoons, martens, dogs, racoon dogs, wolves, badgers, minks, polar fox, otter and red panda), a diverse organ panel (brain, lung, liver, lymph node, kidney and spleen) and geographically widely distributed sampling sites (Germany, Austria, Italy, Spain, Romania, Ukraine and Africa). The primer and probes of both new RT-qPCR systems were located on the *P gene*, whereas the reference method by ELIA and co-workers was located on the *N gene*, which can be a strong explanation for the similar analytical and diagnostic validation outcome of CDV-Mix 3 and CDV-Mix 7.

Additionally, both CDV-RT-qPCR assays can be mutually combined with a heterologous (EGFP-Mix 2-HEX) and an endogenous IC (ß-Actin-Mix 2-HEX) as it is a verification tool for an efficient RNA extraction, a functioning amplification during the PCR run and the presence of inhibitors [31]. ICs are considered a quality criterion in molecular diagnostic methods, especially when using biological sample material of decomposed quality [29,32].

The parallel application of at least two RT-qPCR assays ensures a safe and reliable CDV diagnosis because two independent results per sample are available within one run, while the master mix of each CDV-RT-qPCR assay was reduced to 12.5 μL and the assays are equipped with ICs for quality assurance. The new molecular detection system for CDV was validated with a comprehensive sample panel, indicating a reliable performance on different sample matrices (organs and host animals) in addition to being based on samples of a geographically widespread distribution within Europe. Most of the accessible samples applied for the validation of the new CDV-RT-qPCR assays were collected in European countries and therefore belonged to the European clusters of the CDV lineages. The results are lacking with regard to addressing information about the applicability and reliability of the new molecular detection system in terms of furthering CDV clusters. Future applications of the validated CDV-RT-qPCR assays will underpin their diagnostic capacity with regard to the evolved genetic dynamic on the sequence level recognized in recent years. Furthermore, the diagnostic performance of the CDV-RT-qPCR assays for CDV lineages that have yet to be tested should be addressed.

## Figures and Tables

**Figure 1 viruses-13-01632-f001:**
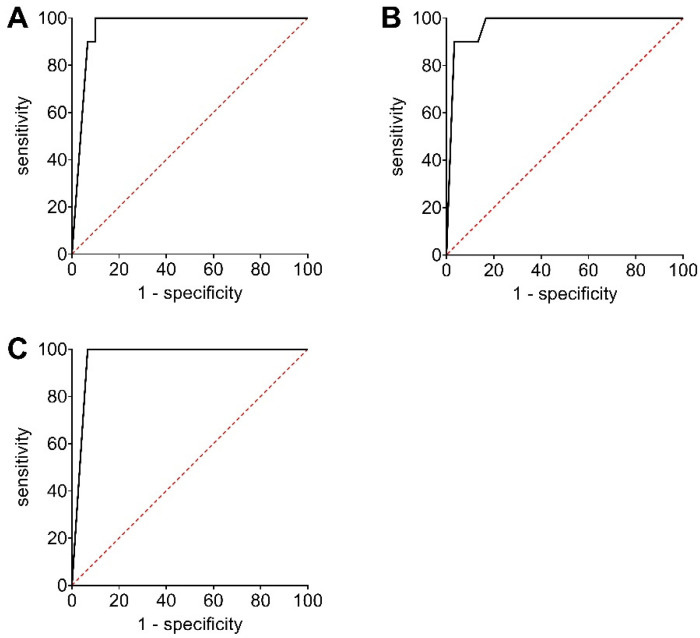
Receiver operating curve (ROC) analyses illustrating the relation of sensitivity and 1-specificity of CDV-Mix 3 (**A**), CDV-Mix 7 (**B**) and ELIA Mix (**C**).

**Table 1 viruses-13-01632-t001:** Sequences of finally used primers and probes.

PCR Assay	Primer/Probe	Sequence 5′–3′	Amplicon Size (bp)	Location	Reference
ELIA-Mix	CDV-F	AGC TAG TTT CAT CTT AAC TAT CAA ATT	87	*N gene*	Elia et al., 2006 [27]
	CDV-R	TTA ACT CTC CAG AAA ACT CAT GC			
	CDV-Pb	FAM-ACC CAA GAG CCG GAT ACA TAG TTT CAA TGC-TAMRA			
CDV-Mix 3	CDV4.1-F	CTG TCR GTA ATC GAG RAT TCG A	116	*P gene*	Scagliarini et al., 2007
	CDV3-R	GCC GAA AGA ATA TCC CCA GTT AG			[28], modified
	CD3.1-FAMas	FAM-ATC TTC GCC AGA RTC YTC AGT GCT-BHQ1			
CDV-Mix 7	CDV-1808-F	AGG ARC AGG CCT AYC ATG TCA	96	*P gene*	in this study
	CDV-1903-R	TRC TGC TGA CCT CTT GAA TCT C			
	CDV-1842-FAM	FAM-ATG CCT CAA ARC CCT CAG AGA GAA TCC-BHQ1			
EGFP-Mix 2-HEX	EGFP-1-F	GAC CAC TAC CAG CAG AAC AC	177	637–794	Hoffmann et al., 2006
	EGFP-10-R	CTT GTA CAG CTC GTC CAT GC			[31]
	EGFP-HEX	HEX-AGC ACC CAG TCC GCC CTG AGC A-BHQ1			
β-Actin-Mix 2-HEX	ACT-1005-F	CAG CAC AAT GAA GAT CAA GAT CAT C	130	1005–1114	Toussaint et al., 2007
	ACT-1135-R	CGG ACT CAT CGT ACT CCT GCT T			[32]
	ACT-1081-HEX	HEX-TCG CTG TCC ACC TTC CAG CAG ATG T-BHQ1			

PCR = polymerase chain reaction, CDV = canine distemper virus, EGFP = enhanced green fluorescent Protein, F = forward primer, R = reverse primer, Pb = probe, bp = base pairs, N = nucleocapsid protein, P = phosphoprotein.

**Table 2 viruses-13-01632-t002:** Evaluation of the analytical sensitivity regarding CDV-Mix 3 and CDV-Mix 7.

Concentration of the Template (in GE/μL)	CDV-Mix 3				CDV-Mix 7			
	No of Positive Detectable Replicates	Mean Cq	SD	CV%	No of Positive Detectable Replicates	Mean Cq	SD	CV%
10^4^	36	24.2	0.29	1.18	36	23.6	0.17	0.70
10^3^	36	27.2	0.14	0.52	36	26.7	0.23	0.87
10^2^	36	30.2	0.16	0.54	36	29.9	0.22	0.74
10^1^	36	33.4	0.31	0.93	36	33.5	0.43	1.29
10^0^	31	37.0	1.11	2.99	13	38.1	1.15	3.02
10^−1^	5	38.7	0.48	1.23	1	38.0	-	-

CDV = canine distemper virus, GE = genome equivalent, Cq = quantification Cycle, SD = standard deviation, CV = coefficient of variation.

## Data Availability

Data are contained within the article and the Appendix A.

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
