# Peer review of "A New Molecular Detection System for Canine Distemper Virus Based on a Double-Check Strategy"

_viruses, 2021, doi:10.3390/v13081632_

Round 1
Reviewer 1 Report
This work developed novel methods for the safe detection of CDV in domestic and wild animals. The new methods achieved a sensitivity of 98.9% and a specificity of 100%, which is a reliable molecular diagnostic tool for the detection of CDV in domestic and wild animals. I have some suggestions shown below:
- In the results section, I highly recommend the authors to show a ROC curve to demonstrate the sensitivity and specificity. A comparison between the new method and previous methods should also be shown in the ROC curves.
- Further explanations are needed to reveal why the new tool can perform much better than previous methods.
- The sensitivity and specificity are extremely high. Besides such high performance, is there any limitation for the new tool?
Reviewer 2 Report
The authors efficiently address the problems generated in diagnostic systems by the genetic variability of CDV and the inconveniences generated by the multispecies potential of CDV. For this reviewer the paper presents an important methodological approach that could be really useful for too many veterinary diagnostic laboratories from different part of the world.
Minor changes
Author should include a paragraph addressing BIAS control in sampling and processing samples. Do the same lab operator perform all the analysis? , Were the Index new rt-qPCR results interpreted without knowledge of the results of the reference standard? , Was there an appropriate interval between index test(s) and reference standard?. For a comprehensive review, I recommend check the QUADAS guide that is designed to assess the quality of primary diagnostic accuracy studies (just as an example of the BIAS report).
Nice paper!
Round 2
Reviewer 1 Report
My questions have been addressed. I recommend it to be accepted.